# Alumina as an Antifungal Agent for *Pinus elliottii* Wood

**DOI:** 10.3390/jof8121299

**Published:** 2022-12-14

**Authors:** Andrey P. Acosta, Ezequiel Gallio, Nidria Cruz, Arthur B. Aramburu, Nayara Lunkes, André L. Missio, Rafael de A. Delucis, Darci A. Gatto

**Affiliations:** 1Postgraduate Program in Mining, Metallurgical and Materials Engineering, Federal University of Rio Grande do Sul, Porto Alegre 90650-001, RS, Brazil; 2Postgraduate Program in Materials Science and Engineering, Federal University of Pelotas, Pelotas 96010-610, RS, Brazil

**Keywords:** softwood, wood-rot fungus, wood protection, Al_2_O_3_, wood treatment, decay fungi

## Abstract

This work deals with the durability of a *Pinus elliotti* wood impregnated with alumina (Al_2_O_3_) particles. The samples were impregnated at three different Al_2_O_3_ weight fractions (c.a. 0.1%, 0.3% and 0.5%) and were then exposed to two wood-rot fungi, namely white-rot fungus (*Trametes versicolor*) and brown-rot fungus (*Gloeophyllum trabeum*). Thermal and chemical characteristics were evaluated by Fourier transform infrared spectroscopy (FT-IR) and thermogravimetric (TG) analyses. The wood which incorporated 0.3 wt% of Al_2_O_3_ presented a weight loss 91.5% smaller than the untreated wood after being exposed to the white-rot fungus. On the other hand, the highest effectiveness against the brown-rot fungus was reached by the wood treated with 5 wt% of Al_2_O_3_, which presented a mass loss 91.6% smaller than that of the untreated pine wood. The Al_2_O_3_-treated woods presented higher antifungal resistances than the untreated ones in a way that: the higher the Al_2_O_3_ content, the higher the thermal stability. In general, the impregnation of the Al_2_O_3_ particles seems to be a promising treatment for wood protection against both studied wood-rot fungi. Additionally, both FT-IR and TG results were valuable tools to ascertain chemical changes ascribed to fungal decay.

## 1. Introduction

In several countries, like Brazil, the United States and Canada, *Pinus elliottii* is one of the most used wood species for some important industrial applications, such as furniture, civil construction, pulp and paper, and resin extraction. Although pine can be considered a high-quality engineering material, its high susceptibility to the attack of xylophagous agents restricts its use in certain applications. Among these microorganisms, fungi are the etiologic agents responsible for the biodegradation of lignocellulosic biomasses since they can access nutrients from the wood cell wall [1].

Among the wood-rotting fungi, those called white-rot and brown-rot fungi can be highlighted since their action may hydrolyze structural compounds from wood, such as cellulose and lignin [2]. White-rot fungi can metabolize cellulose, hemicellulose and lignin [3,4], whereas brown-rot fungi may consume polysaccharides rather than lignin [5]. Usually, wood exposed to a white-rot fungus progressively loses strength. In this sense, Witomski et al. [6] showed that both flexural and compressive strengths of a Scots pine wood decreased by up to 50% after standard decaying tests were performed using a white-rot fungus. Among the white-rot fungi, *Trametes versicolor* stands out since its enzymes, like laccase and lignin peroxidase, may destroy most lignified plants, such as wood and natural fibers [7,8]. Regarding brown-rot fungi, a *Gloeophyllum trabeum* infestation may be identified by quick and extensive depolymerization of cellulose chains, leading to high strength losses even when associated with low weight losses [9,10]. For instance, Calonego et al. [11] reported mass losses between 1.97% and 12.2% for an *Eucalyptus grandis* wood decayed by a *G. trabeum* fungus after a standard biological assay.

The impregnation of inorganic particles, endowed with high anti-fungal activities, is one of the main ways to improve wood durability [12]. Regarding the literature, several promising results were already ascribed to the incorporation of inorganic particles in the wood [13,14], including silver nitrate (AgNO_3_) [15], zinc oxide (ZnO), and titanium dioxide (TiO_2_) [16] among others [2]. For instance, Nair et al. [17] impregnated a *Hevea brasiliensis* wood with two inorganic nanoparticles, namely ZnO and copper (II) oxide (CuO), which were dispersed within a propylene glycol solution. Based on comparisons with their untreated wood, they reported that the treatment yielded decreases in mass loss of 35% and 50% for woods attacked by white-rot and brown-rot fungi, respectively. De Filpo et al. [18] impregnated TiO_2_ particles in six different wood species and exposed them to white-rot and brown-rot fungi. They obtained decreases of up to 50% in fungus spreading and attributed these results to the presence of these inorganic particles on the wood cell wall.

Al_2_O_3_ particles have been widely used for different purposes, such as adsorbing substances and catalysts [19]. Besides that, these particles have high levels of chemical and thermal stabilities and mechanical strength, as well as high thermal and electrical insulation properties and availability [20]. Additionally, recent studies on Al_2_O_3_-treated woods showed that these particles have a high antibacterial activity [21] and may form a physical barrier that may cover the wood, conferring protection against moisture and xylophagous agents [1].

Changes in chemical composition, attributed to wood biodegradation, can be accessed by traditional quantitative analyses, which involve previous milling of the samples. Nevertheless, there are non-destructive methods which may be advantageous in terms of cost, speed and even reliability. In this sense, satisfactory results obtained through Fourier transform infrared spectroscopy (FT-IR) were recently published for chemical modifications caused by wood-rot fungi [22,23]. In these studies, the spectra can be considered a qualitative result, although there are mathematical approaches to measure some peak intensities and create quantitative data. Besides the FT-IR, a thermogravimetric (TG) analysis can be considered a complementary technique [24,25,26]. In woods, the time for heating over causes noticeable mass losses. This can be attributed to the release of moisture and volatile organic matter, as well as the decomposition of polysaccharides (c.a. hemicelluloses and cellulose) and lignin [27]. Therefore, this technique may bring valuable results for studies on chemical changes in wood ascribed to its biodegradation.

This study aimed at verifying the effectiveness against two wood-rot fungi of Al_2_O_3_ particles. These were impregnated at different weight fractions into a *P. elliottii* wood, using both FT-IR and TG data.

## 2. Materials and Methods

### 2.1. Material Selection and Wood Treatment

Sapwood specimens were obtained from young *Pinus elliottii Engelm* (P_E_) trees (approximately 30 years old). Prismatic samples were cut with the dimensions of 15 × 15 × 9 mm^3^ (radial × tangential × axial) and were then placed into a climatic chamber at 20 °C and 65% relative humidity (RH) until reaching weight stability. Alumina (Al_2_O_3_) particles were obtained by a calcination process at 1000 °C, involving NH_2_CONH_2_ (urea) and Al(NO_3_)_3_ (aluminium nitrate) at a proportion of 2.5:1. Four different solutions were prepared using three different Al_2_O_3_ weight fractions (c.a. 0.1%, 0.3% and 0.5%). For that, the Al_2_O_3_ particles were dispersed in an aqueous solution of sodium polyacrylate (SPAC at a concentration of 5 wt% in relation to the Al_2_O_3_ weight). The SPAC had a molecular weight of 5100 g·mol^−1^ and played a role in keeping the particles homogeneously dispersed in the water [28,29].

The impregnation process occurred through the full-cell process (Bethell) in a laboratory autoclave, wherein an initial vacuum was applied for 15 min in order to remove the entrapped air within both the wood and pressure chamber. The treatment solution was introduced, taking advantage of the previous vacuum, and then a positive pressure of 8 kgf·cm^−2^ was applied for 90 min. Afterwards, the samples were stored in the aforementioned climatic chamber until reaching stable mass again.

### 2.2. Decay Resistance Tests

Wood-rot fungi resistance was evaluated for six samples per group according to ENV-12038 standard with some adaptations regarding some glassware and the BOD clarity. The samples were oven-dried at 100 °C until reaching constant mass and then autoclaved at 120 °C for 20 min together with the glassware. A potato dextrose malt agar, used as a culture medium, was placed on Petri dishes, which were then placed into a BOD chamber at 23 °C and 65–70% RH for 15 days. Afterwards, the dishes with any contamination were discarded. Those uncontaminated glasses were inoculated with two 2 cm discs of *Trametes versicolor* and *Gloeophyllum trabeum* fungi and placed again in the BOD chamber, where they remained for 16 weeks. These fungi are extensively mentioned in standard procedures and research articles [30,31,32,33,34]. The specimens were then cleaned by brushing to remove all fungi mycelium, dried at 103 °C for 24 h and weighed to obtain the mass loss according to Equation (1), which is described in ASTM D2017. All mass measurements were performed using an analytical scale with a 0.001 g resolution.
(1)$$\mathrm{ML}\;(\%)=\left(\frac{M_{I}-M_{F}}{M_{I}}\right)\times 100$$

### 2.3. FT-IR and TG Analyses

Chemical changes attributed to the biodegradation processes were analyzed by Fourier transform infrared spectroscopy (FTIR) and thermogravimetric (TG) analyses. The FT-IR spectra were conducted on solid samples using an FT-IR Jasco 4100 spectrometer (Jasco, Tokyo, Japan) with a resolution of 4 cm^−1^. Each spectrum was the average of 32 scans recorded at the 600–2000 cm^−1^ wavenumber range. The resulting spectra were normalized according to the band at 1030 cm^−1^, which is common for wood samples. After that, quantitative data were obtained based on the intensities of bands attributed to carbohydrates and lignin. The lignin was evaluated through the band at 1509 cm^−1^, which is exclusively related to aromatic skeletal vibration in this molecule [22,35]. On the other hand, the wood carbohydrates were represented by the bands at 890 cm^−1^, 1370 cm^−1^, 1420 cm^−1^ and 1740 cm^−1^ [1,22,36]. Furthermore, chemical changes caused by the decaying process, which also changed the thermal stability, were analyzed by thermogravimetric (TG) analysis. For that, a Navas TGA 1000 equipment was adjusted to a heating rate of 10 °C·min^−1^ from room temperature (c.a. 20 °C) to 600 °C under an N_2_ atmosphere.

### 2.4. Statistical Analysis

The numeric data were subjected to the Levene and Shapiro–Wilk tests to ascertain the homogeneity of variances and the normality of data, respectively. Then, one-way analysis of variance (ANOVA) and Fisher tests, both at 5% significance, were performed to compare the means of the groups. Finally, the means of I_Lignin_/I_carbohydrate_ ratio were compared using Snedecor F tests, implemented at 5% significance.

## 3. Results and Discussion

### 3.1. Mass Loss Results

Figure 1 shows the mass loss data for the treated and untreated *Pinus elliottii* woods caused by fungi exposure. Regarding the *G. trabeum* fungus (Figure 1A), only the treatment with 0.5% of Al_2_O_3_ yielded a significantly smaller mass loss in relation to the untreated wood. This difference was 91.5% in percentage values. On the other hand, still compared to the untreated wood, all treated woods presented significantly smaller mass losses. This was attributed to the *T. versicolor* fungus (Figure 1B), and the treatment with 0.3% of Al_2_O_3_ stood out with a decrease of 91.6% in this comparison. These different biodegradation mechanisms are associated with different opposing and synergistic mechanisms that occur during a biological assay. For instance, Shang et al. [1] studied a *Betula platyphylla* wood, which was exposed to four wood-rot fungi, and observed mass losses varying in the 34.97–62.55% range. In general, white-rot fungi preferably consume compounds from lignin, whereas brown-rot fungi usually attack all major wood compounds [37]. Besides that, the difference between the mass losses of the woods, treated with 0.3% and 0.5% of Al_2_O_3_ and exposed to the *G. trabeum* fungus, may be attributed to an uneven distribution of these particles on the wood surface. This contrasted with an even distribution in opened spaces from large anatomical elements, like tracheids.

In general, the wood protection imparted by the treatments can be attributed to the formation of a physical barrier upon the wood surface, which probably restricted the natural spreading of the fungi. Shang et al. [1] affirmed that, in addition to the difficulty in developing normally, fungi that infest one wood treated by particle impregnation have difficultly accessing the moisture from this wood. Similarly, Ghorbani et al. [13] treated a *Populus deltoides* wood by impregnation of silica nanoparticles and found mass losses from 1.85% to 29.00% after inoculating a *T. versicolor* fungus. These authors also attributed this increase in resistance against fungi to the formation of a barrier of nanoparticles that hindered the access of moisture and nutrient reserves from the wood cell wall.

### 3.2. Chemical Changes Accessed by FT-IR

As commented above, the *T. versicolor* fungus may indistinctly degrade all wood components, whereas the *G. trabeum* fungus preferably degrades hemicelluloses and cellulose rather than lignin [5,38,39]. Therefore, in a comparison with an undecayed sample, one signal detected by FT-IR analysis for a decayed sample represents a chemical modification which is attributed to the biodegradation process [40]. Fungi consume wood by secreting different enzymes. These are able to convert macromolecules (c.a. cellulose, hemicelluloses and lignin) into low molecular weight compounds via a process called enzymatic hydrolysis [41,42]. Therefore, it is expected that some new functional chemical groups are represented by new prominent bands (e.g., 720 cm^−1^ and 750 cm^−1^) in the FT-IR spectrum [5].

Figure 2A shows the FT-IR spectra for the untreated *Pinus elliottii* woods exposed to the studied fungi. Compared to the undecayed wood, bands associated with hemicelluloses at 1725 cm^−1^ and 670 cm^−1^ disappeared for the wood exposed to the *G. trabeum* fungus and, in addition, the band at 810 cm^−1^ (related to lignin) looked like attenuated [22]. Additionally, in this same comparison, a band appeared at 750 cm^−1^ and another one shifted from 890 cm^−1^ to 860 cm^−1^. In addition, changes in peak intensities at 1640 cm^−1^ (lignin-related compounds) and 1370 cm^−1^ (carbohydrates-related compounds) were attributed to the degradation processes of both two fungi [43]. Finally, a new band appeared at 720 cm^−1^. Regarding the *T. versicolor* fungus, changes in lignin-related and polysaccharides-related compounds were associated with attenuations in bands at 1509 cm^−1^ and 810 cm^−1^ [44], and 670 cm^−1^ [44], respectively. Modified peak intensities at 1460 cm^−1^, 1420 cm^−1^ and 1260 cm^−1^ were also found for untreated woods attacked by both fungi, which may mean changes in cellulose, hemicelluloses and lignin contents [45].

Once both lignin and carbohydrates can be degraded in a durability test, a decrease in the I_Lignin_/I_carbohydrate_ ratio indicates degradation in lignin-related compounds. Conversely, an increase in this property is associated with deterioration mechanisms in wood carbohydrates (Figure 2B). Regardless of the fungus type, the I_Lignin_/I_carbohydrate_ ratio was significantly impacted by the fungal attack. Compared to the unaged wood, that wood exposed to the *G. trabeum* fungus showed a decrease in the following ratios: I_1509/I890_, I_1509/I1370_, I_1509/I1420_ and I_1509/I1740_. This indicates that the lignin was more degraded than the carbohydrates. This can be explained by the enzyme selectivity of brown-rot fungi since carbohydrates are prone to become cleavage sites for their hydrolysis processes, although these fungi may degrade all the major wood compounds if there is time enough. In addition, the band at 1509 cm^−1^ seems to be unchanged, even after the low degradation of the lignin. On the other hand, the wood attacked by the *T. versicolor* fungus presented an increase in the I_1509_/I_890_ ratio accompanied by decreases in the I_1509_/I_1370_, I_1509_/I_1420_ and I_1509_/I_1740_ ratios, which indicates that the wood carbohydrates are prone to being hydrolyzed by the enzymes secreted from this fungus.

Compared to its respective untreated wood decayed by the *T. versicolor* fungus, the wood treated with Al_2_O_3_ presented an attenuated band at 1509 cm^−1^, which was visualized for all Al_2_O_3_ weight fractions. This result indicates the degradation of lignin-related compounds. Also compared to its respective untreated wood, some bands at 1365 cm^−1^ (C-H deformation in carbohydrate-related compounds [5]) and 670 cm^−1^ (OH bending and out-of-plane stretching of hemicellulose-related compounds [1]) disappeared for the wood treated with 0.1% of Al_2_O_3_ exposed to the *T. versicolor* fungus (Figure 3). Regarding the woods attacked by the *G. trabeum fungus*, compared to the undecayed wood, all decayed ones presented the band at 810 cm^−1^ shifted to 770 cm^−1^. This may be associated with a chemical modification of lignin-related molecules. In this same comparison, the attenuated bands at 1425 cm^−1^ and 1370 cm^−1^ are related to degradation mechanisms in carbohydrates and lignin, respectively [46], which was already associated with increases in lignin content [23].

The decreases in the I_1509_/I_890_, I_1509_/I_1370_ and I_1509_/I_1420_ ratios indicated that the lignin of the wood treated with 0.1% of Al_2_O_3_ was more degraded than the respective carbohydrates. However, the increase in the I_1509_/I_1740_ ratio of the wood degraded by the *G. trabeum* fungus also suggests that the lignin was significantly attacked. The wood which was treated with 0.3% of Al_2_O_3_ presented an attenuated band at 1509 cm^−1^ (lignin-related compounds) after the degradation of both fungi (Figure 4). Furthermore, attenuations in the bands located at 1320 cm^−1^ and 1260 cm^−1^ are commonly attributed to degradation mechanisms in cellulose, hemicelluloses and syringyl lignin-related compounds caused by the attack of wood-rot fungi [42].

The band at 1740 cm^−1^ disappeared after exposure to the *G. trabeum* fungus. This band is related to hemicelluloses-related compounds [46], and its disappearance indicates the conversion of their cyclic structures into chromophoric substances [47] due to the action of the enzymes secreted by xylophagous fungi [5,39]. In this sense, Tomak et al. [23] found strong correlations between mass loss and the intensity of the band at 1740 cm^−1^. Regarding the I_Lignin_/I_carbohydrate_ ratios, again it seems that the fungi preferably consumed lignin-related compounds compared to carbohydrate-related ones.

It is possible to verify, for the wood impregnated with 0.5% of Al_2_O_3_, that the bands at 1700 cm^−1^ and 1460 cm^−1^ disappeared in relation to the untreated wood and, in addition, there was an increase in the peak intensity of the band at 1640 cm^−1^ (related to lignin-related compounds and chemically linked water) [46] (Figure 5). This result is likely related carbonyl groups derived from the lignin degradation and the overall adsorption of water [22,39]. Additionally, there is a mechanism of wood weathering in which the water movement on the wood surface lixiviates some small molecules generated by the degradation of lignin and carbohydrates. After that, new wood layers are exposed to a new cycle of fungal infestation. The attack of the G. trabeum fungus also yielded an increase in the band at 1150 cm^−1^ (vibration of C−O−C bonds from cellulose and hemicelluloses [22]), accompanied by the disappearance of the band at 890 cm^−1^ (out-of-plane bending of the exomethylene groups from cellulose and hemicelluloses [46]).

According to Darwish et al. [48], a decrease in the band at 1460 cm^−1^ indicates decreases in cellulose and lignin contents due to the depolymerization processes of these macromolecules. The same can be stated about the bands at 1030 cm^−1^ and 890 cm^−1^ [40]. All these spectrum signals are associated with the enzymatic hydrolysis of glucopyranose rings, yielding the release of aldehyde groups, which can be detected by FT-IR [48]. Regarding the I_Lignin_/I_carbohydrate_ ratios of the woods treated with 0.5% of Al_2_O_3_, both fungi attacks resulted in significant increases, indicating a preference for carbohydrate degradation over lignin.

### 3.3. Chemical Changes Accessed by TG

Figure 6 shows the TG curves of all treated and untreated woods before and after the fungal attack. All samples presented low mass losses in the 30–105 °C region, which are related in all cases to the desorption of moisture. The undecayed woods presented clearly higher mass losses in this region, a fact which indicates that the wood biodegradation decreased their equilibrium moisture content. This progressive mass loss was maintained during the second step, which occurred until approximately 300 °C and was related to the decomposition of hemicelluloses and amorphous cellulose. Compared to each respective undecayed wood, the decayed woods presented higher mass losses in the 200–300 °C temperature range, since amorphous carbohydrates are prone to be hydrolyzed by fungal enzymes. All the decayed woods presented higher mass losses compared to the undecayed ones in the 300–400 °C temperature range, which is probably due to the incorporation of the fungi inside and onto the wood cell wall.

The decomposition of the lignin started above 400 °C and, in this region, no different TG patterns were found in a comparison between all samples. This occurred due to the low-weight fractions of Al_2_O_3_ being incorporated into the woods. Therefore, the known high thermal stability of the powdered Al_2_O_3_ was not transmitted to the treated woods since the Al_2_O_3_ contents were not high enough. Compared to the respective undecayed woods, slightly higher mass losses above 450 °C were found for the woods decayed by the *T. versicolor* fungus, which again can be explained by the aforementioned feed preference of this fungus.

Compared to the untreated wood, the undecayed treated ones presented slight increases in thermal stability (Table 1). Examples included all undecayed treated woods for the 105–400 °C temperature range, as well as the woods treated with 0.3% and 0.5% of Al_2_O_3_ and decayed by *T. versicolor* for the 105–200 °C and 200–300 °C, as well as these same woods decayed by *G. Trabeum* for the 300–400 °C temperature range. Furthermore, increases in thermal stability for the 105–200 °C and above 600 °C were also found in all the Al_2_O_3_-treated woods decayed by *G. Trabeum*. The increases in thermal stabilities for the 105–300 °C can be considered to be outstanding results since they indicate that the treatments protected some thermally unstable wood compounds, like amorphous carbohydrates. Some authors already stated that the most biologically vulnerable wood compounds are those amorphous carbohydrates, which are rich in thermally unstable compounds due to their high amount of side secondary bonds [49,50]. Several recently published studies portrayed the application of conventional wood treatments. This yielded increases in thermal stability due to previous degradations of these thermally unstable compounds [50]. For instance, Liu et al. [51] carried out a heat treatment at 120 °C for 4 h on a *Populus tomentosa* wood and found an improved thermal stability, which could be ascribed to the degradation of side groups from hemicelluloses.

The T_ONSET_ and T_ENDSET_ (temperatures related to initial and final mass losses, respectively), as well as the T_MAX_ (maximum temperature), were determined for the studied samples (shown in Figure 6). The fungal attack yielded decreases in thermal stability based on these properties. This effect was more prominent for the wood decayed by the *T. versicolor* fungus, which is probably due to the severe attack of the lignin in this case since lignin is the most thermally stable wood macromolecule [49]. The woods decayed by the *T. versicolor* fungus presented the most severe mass losses (Figure 2), which explains their smallest levels of T_MAX_. Additionally, all woods decayed by this fungus presented decreased levels of T_ONSET_, T_ENDSET_ and T_MAX_ in lower temperature ranges.

The increase in thermal stability above 600 °C, found in all the Al_2_O_3_-treated woods decayed by *G. Trabeum*, can be attributed to the a lower lignin degradation caused by this basidiomycetes fungus, as aforementioned. In this sense, it is known that lignin plays a crucial role in several biological and thermal properties of wood, as its highly aromatic structure confers high thermal stability and high resistance against xylophagous fungi [52]. Finally, it appears that the FT-IR and TG techniques can be used in a complementary way for the analysis of woods subjected to biodegradation mechanisms caused by wood-rot fungi, since these microorganisms cause chemical and thermal stability changes detectable by these techniques. In addition, numerical data, like I_Lignin_/I_Carbohydrate_ ratio, T_ONSET_, T_ENDSET_ and T_MAX_, bring greater precision and depth to these analyses.

## 4. Conclusions

The impregnation of different weight fractions (0.1%, 0.3% and 0.5%) of Al_2_O_3_ resulted in slight improvements in thermal stability. This indicates that the physical barrier formed by the impregnated particles restricted the hyphal spreading growth and protected some of the major wood components, such as hemicelluloses and cellulose. The Al_2_O_3_-treated woods presented different FT-IR signals at 670, 1150, 1460 and 1700 cm^−1^, indicating that there were chemical changes attributed to the treatments. Based on the mass loss data, which indicated the wood durability, the impregnation of the Al_2_O_3_ particles conferred protection to the studied pine wood against the attack of both *T. versicolor* and *G. trabeum* fungi, especially the treatments adjusted at 0.3% and 0.5% Al_2_O_3_ content. Changes in chemical composition were associated with attenuated or increased FT-IR peak intensities, as well as either the disappearance or emergence of new bands. Both applied analyses (namely FT-IR and TG) proved to be valuable tools to understand the influence of the studied decay mechanisms.

## Figures and Tables

**Figure 1 jof-08-01299-f001:**
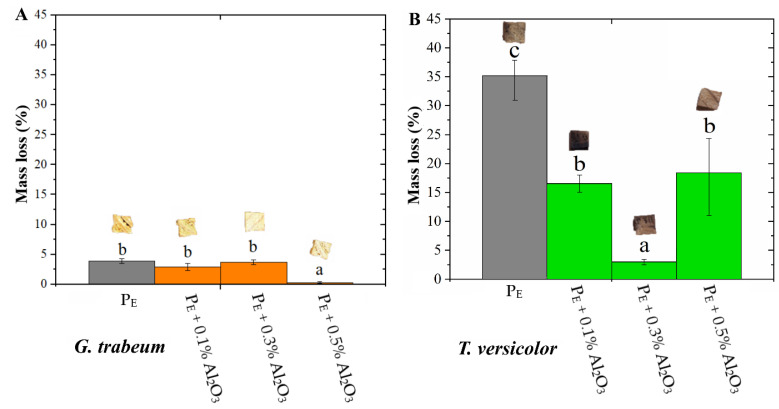
Mass losses of the treated and untreated *P. elliottii* (P_E_) woods exposed to the *G. trabeum* (**A**) and *T. versicolor* (**B**) fungi, where different letters above the bars indicate statistically different means.

**Figure 2 jof-08-01299-f002:**
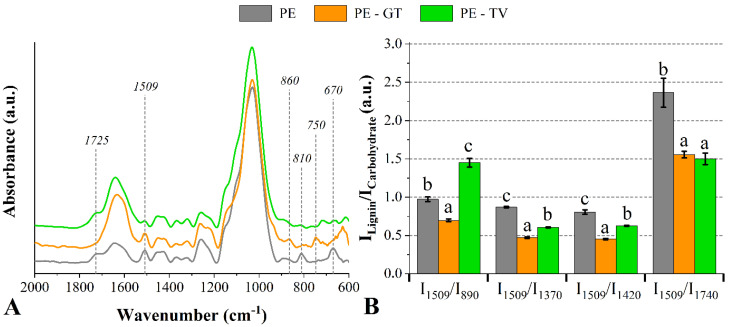
FT−IR spectra (**A**) and lignin/carbohydrate ratios (**B**) of the untreated *P. elliottii* (P_E_) woods exposed to the *G. trabeum* and *T. versicolor* fungi. Where: Different letters above the bars indicate statistically different means for each lignin/carbohydrate ratio.

**Figure 3 jof-08-01299-f003:**
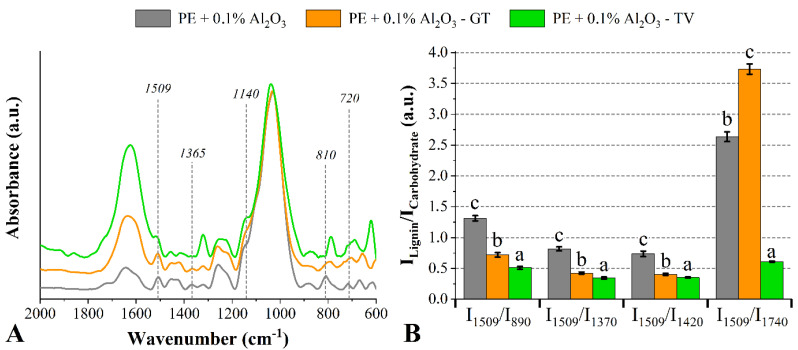
FT−IR spectra (**A**) and lignin/carbohydrate ratios (**B**) of the *P. elliottii* (P_E_) woods treated with 0.1% of Al_2_O_3_ exposed to the *G. trabeum* and *T. versicolor* fungi. Where: Different letters above the bars indicate statistically different means for each lignin/carbohydrate ratio.

**Figure 4 jof-08-01299-f004:**
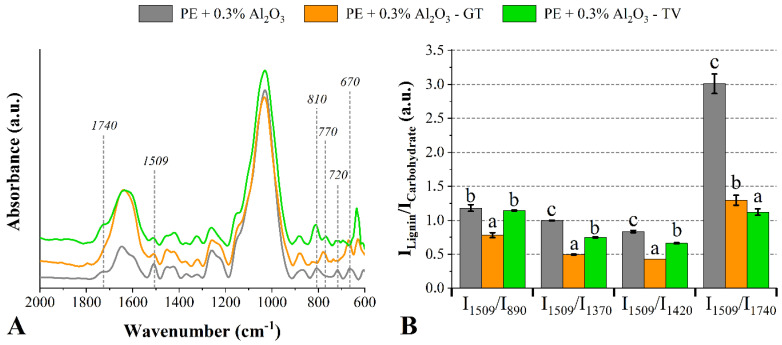
FT−IR spectra (**A**) and lignin/carbohydrate ratios (**B**) of the *P. elliottii* (P_E_) woods treated with 0.3% of Al_2_O_3_ exposed to the *G. trabeum* and *T. versicolor* fungi. Where: Different letters above the bars indicate statistically different means for each lignin/carbohydrate ratio.

**Figure 5 jof-08-01299-f005:**
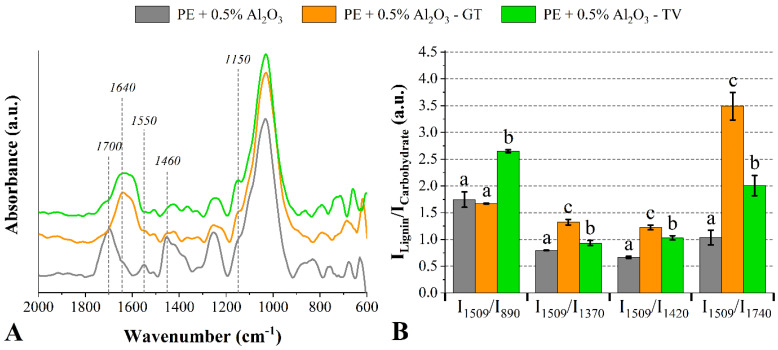
FT−IR spectra (**A**) and lignin/carbohydrate ratios (**B**) of the *P. elliottii* (P_E_) woods treated with 0.5% of Al_2_O_3_ exposed to the *G. trabeum* and *T. versicolor* fungi. Where: Different letters above the bars indicate statistically different means for each lignin/carbohydrate ratio.

**Figure 6 jof-08-01299-f006:**
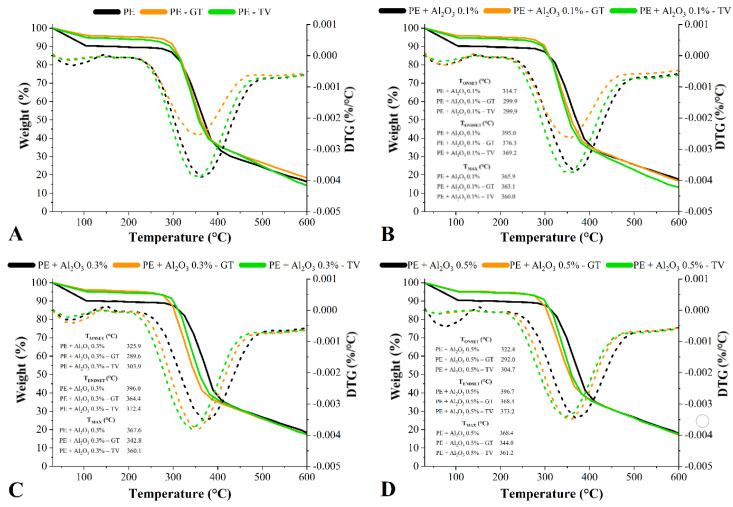
TG curves and main thermal parameters of the treated and untreated *P. elliottii* (P_E_) woods exposed to the *G. trabeum* (**A**,**C**) and *T. versicolor* (**B**,**D**) fungi.

**Table 1 jof-08-01299-t001:** Mass losses for different temperature ranges of the treated and untreated *P. elliottii* (P_E_) woods exposed to the *G. trabeum* (GT) and *T. versicolor* (TV) fungi.

Treatment	Mass Loss (%)					>600 (°C)
	30–105 (°C)	105–200 (°C)	200–300 (°C)	300–400 (°C)	400–600 (°C)	
P_E_	9.75	0.78	3.98	50.67	18.49	16.33
P_E_ − GT	3.91	0.87	5.40	53.60	17.86	18.36
P_E_ − TV	5.09	0.94	6.43	51.73	21.58	14.23
P_E_ + Al_2_O_3_ 0.1%	9.79	0.50	2.75	50.24	19.15	17.57
P_E_ + Al_2_O_3_ 0.1% − GT	4.24	0.75	5.39	54.24	18.64	16.74
P_E_ + Al_2_O_3_ 0.1% − TV	5.31	0.51	6.44	53.80	20.75	13.19
P_E_ + Al_2_O_3_ 0.3%	9.83	0.70	2.08	48.96	20.05	18.38
P_E_ + Al_2_O_3_ 0.3% − GT	4.06	0.53	9.54	50.89	17.66	17.32
P_E_ + Al_2_O_3_ 0.3% − TV	4.83	0.79	4.36	54.00	18.55	17.47
P_E_ + Al_2_O_3_ 0.5%	9.53	0.76	2.13	49.15	20.54	17.89
P_E_ + Al_2_O_3_ 0.5% − GT	4.64	0.55	7.91	50.68	19.09	17.13
P_E_ + Al_2_O_3_ 0.5% − TV	4.88	0.51	4.79	53.60	18.55	17.67

## Data Availability

The study did not report any data.

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
