# Peer review of "Alumina as an Antifungal Agent for Pinus elliottii Wood"

_jof, 2022, doi:10.3390/jof8121299_

Round 1

Reviewer 1 Report

The article „Alumina as an anti-fungal agent for Pinus elliottii wood” presents the results of the study on biological effectiveness of alumina used for wood impregnation against selected brown- and white-rot fungi. The research is interesting and important, and several experiments have been performed, including fungal tests, FT-IR, and TGA analyses. However, the article lacks a scientific discussion of the obtained results, several statements contradict one another, and many more or less serious mistakes can be found in the text. Therefore, I cannot recommend the manuscript for publication in this form and suggest its thorough correction and supplementation with relevant information and a good discussion. More detailed comments, suggestions, and questions can be found in the attached pdf file.

Author Response

REBUTTAL TO REFEREES' REMARKS

The authors would like to thank the reviewers for their valuable contribution to the manuscript. Our answers are listed below. All suggestions were accepted and the manuscript was revised accordingly. The changes were highlighted in blue colour.

Check the corrections in the attached file.

Reviewer 2 Report

1)      Title: Ok

2)    Authors fellow the Fungi template 

3)      Abstract:  English check is mandatory

4)      Keywords: you can add one more keyword

5)      Add list of abbreviation and nomenclature

6)      Introduction: please add references literature studies of previous works on the same subject to show clearly the state of the art of your work.

7)      Which size you can consider your approach is efficient.

8)      Results need more discussions emphasize the advantages of your stydies 

9)  References: fellow the Fungi template and update them by adding 2023, With regards

Author Response

(The authors gave the same response as above.)

Reviewer 3 Report

The authors of the manuscript ”Alumina as an anti-fungal agent for Pinus elliottii wood” present studies aimed to verify alumina (Al2O3) as protecting agent againt fungal degradation. FTIR and TGA analysis were carried out for evaluatin chemical composition changes of plant cell wall. The manuscript is clearly written and the results are of interest but, it contains some flawn to be improved before acceptance.

Points that need to be addressed.

ABSTRACT

Line 15: Please, use full name (Trametes versicolor), it is first time cited.

Line 17: Please, use full name, it is first time cited.

Line 21: Please, use thermogravimetric analysis instead TGA.

 INTRODUCTION

Line 72: Please, use P. elliottii instead of "Pinus elliottii".

Line 73: Please, use T. versicolor instead of "Trametes versicolor"

Line 74: Please, use G trabeum instead of “Gloeophyllum trabeum

Materials and Methods

Line 94: Authors may delete table 1, paragraph is ok.

 RESULT AND DISCUSSION

Line 148: Please, if you put panels on figures, cite them on text as Figure 1A or Figure 1B.

Line 148: Please, use P. elliottii instead of "Pinus elliottii".

Lines 154-158: This is a long sentence apparently there is a problem at line 155 at least. Please check.

Line 156: Authors should write “Betula platyphylla” in italic.

Line 160: Please, authors should write “brown rot fungi”.

Figure 1: Please, change colors on bars. This color scale doesn't help. Try to use same color for same alumina % in both panels. About panel capital letters, probably on panel top it is better.

Legend Figure 1: Please use T. versicolor and G. trabeum instead the full names.

Line 165-167: If this is true, why in figure 1B (T. versicolor) does the concentration of Al2O3 at 0.3% show a lower mass loss than the concentration of 0.5%? Is it only a problem of this biological system? Please discuss it a bit.

Line 167 and 169/170: Please, use the proper reference format (Journal of fungi).

Lines 165-174: Please, authors should better connect the first sentence (lines 165-167) with the second part of the paragraph (Shang et al...). Maybe, they should re-write it.

Line 187: Probably, it is better to use Figure 2A.

Line 195: Authors should write “T. versicolor” in italic.

Legend Figure 2: Please, use T. versicolor, G. trabeum and P. elliottii instead of the full name.

Line 203: Authors should indicate "Figure 2B" somewhere on text.

Line 210: Authors should write “T. versicolor” in italic.

Line 213: Authors should write “P. elliottii” in italic.

Line 218: Authors should write “T. versicolor” in italic.

Legend Figure 3: Please, write “T. versicolor” and “G. trabeum” in italic and write “P. elliottii” instead of the full name.

Line 236: Authors should write “T. versicolor” in italic.

Legend Figures 4,5,6,7,8,9: See legend Figure 3.

CONCLUSION

Line 387: Please, write “P. elliottii”  instead "Pinus elliottii".

Line 391-393: Please, correct the last part of the sentence.

Author Response

(The authors gave the same response as above.)

Round 2

Reviewer 1 Report

The manuscript has been corrected according to comments and suggestions. Therefore, I recommend it for publishing.